# Stimuli-Responsive Drug Delivery of Doxorubicin Using Magnetic Nanoparticle Conjugated Poly(ethylene glycol)-*g*-Chitosan Copolymer

**DOI:** 10.3390/ijms222313169

**Published:** 2021-12-06

**Authors:** Hyun-Min Yoon, Min-Su Kang, Go-Eun Choi, Young-Joon Kim, Chang-Hyu Bae, Young-Bob Yu, Young-IL Jeong

**Affiliations:** 1Department of Industrial and Management Engineering, POSTECH, Gyeongbuk, Pohang 37673, Korea; yhm0610@naver.com; 2Department of Bio and Brain Engineering, KAIST, Daejeon 34141, Korea; nsa29213@gmail.com; 3College of Medicine, Hanyang University, Seoul 04763, Korea; swan1917@naver.com; 4Gwangju Science Academy, Gwangju 61005, Korea; biokimyj@hanmail.net; 5Department of Well-being Bioresources, Sunchon National University, Suncheon 57922, Korea; chbae@scnu.ac.kr; 6Department of Emergency Medical Rescue & Department of Herbal Pharmaceutical Development, Nambu University, Gwangju 62271, Korea; 7The Institute of Dental Science, Chosun University, Gwangju 61452, Korea; 8Research Institute of Convergence of Biomedical Science and Technology, Pusan National University Yangsan Hospital, Gyeongnam, Yangsan 50612, Korea

**Keywords:** iron-oxide nanoparticle, magnetic nanoparticle, redox-responsiveness, chitosan, stimuli-responsiveness

## Abstract

Stimuli-responsive nanoparticles are regarded as an ideal candidate for anticancer drug targeting. We synthesized glutathione (GSH) and magnetic-sensitive nanocomposites for a dual-targeting strategy. To achieve this goal, methoxy poly (ethylene glycol) (MePEG) was grafted to water-soluble chitosan (abbreviated as ChitoPEG). Then doxorubicin (DOX) was conjugated to the backbone of chitosan via disulfide linkage. Iron oxide (IO) magnetic nanoparticles were also conjugated to the backbone of chitosan to provide magnetic sensitivity. In morphological observation, images from a transmission electron microscope (TEM) showed that IO nanoparticles were embedded in the ChitoPEG/DOX/IO nanocomposites. In a drug release study, GSH addition accelerated DOX release rate from nanocomposites, indicating that nanocomposites have redox-responsiveness. Furthermore, external magnetic stimulus concentrated nanocomposites in the magnetic field and then provided efficient internalization of nanocomposites into cancer cells in cell culture experiments. In an animal study with CT26 cell-bearing mice, nanocomposites showed superior magnetic sensitivity and then preferentially targeted tumor tissues in the field of external magnetic stimulus. Nanocomposites composed of ChitoPEG/DOX/IO nanoparticle conjugates have excellent anticancer drug targeting properties.

## 1. Introduction

Colloidal-based drug delivery vehicles such as core-shell-type nanoparticles, polymeric micelles, and nano-conjugates have been extensively investigated to achieve site-specific drug delivery for the treatment of diseased cells or organs [1,2,3,4,5]. Due to their small diameter, nanoparticles can be administered intravenously and then delivered to specific organs, tissues, or cells [3,4,5]. Furthermore, nanoparticles are frequently employed to improve solubility of hydrophobic anticancer agents and to prolong blood circulation time [1,2,3]. However, nanoparticles still have limits to achieving efficient delivery of the drug to the target site; i.e., administered nanoparticles normally spread out in the body even though their delivery capacity to the target site is higher than conventional formulations. From these points of view, stimuli-sensitive nanoparticles have been spotlighted in recent decades because they can be concentrated in the target site through physiological, chemical, or physical stimuli such as temperature, pH, magnetic, ultrasound, and light [6,7,8,9,10]. Among them, iron oxide (IO) nanoparticles have been widely used to achieve these goals; i.e., IO nanoparticles respond to the magnetic field and then lead to apoptosis of cancer cells [10,11]. Guo et al. reported that an external magnetic field enhances cellular internalization and tumor accumulation of IO nanoparticles [12]. Furthermore, they argued that the tumor-targeting efficiency of IO nanoparticles was varied according to the diameter [12]. Other researchers also reported the synthesis of Fe_3_O_4_-decorated Cu_9_S_5_@-mSiO_2_@Fe_3_O_4_-PEG nanocomposites and external magnetic stimulus induces synergistic therapeutic effect against H22 tumor-bearing animal study [13].

Chitosan, a natural polysaccharide, has been investigated extensively in the biomedical field due to its biocompatibility and functionality [14,15,16]. Specifically, chitosan has an amine group in the backbone and, therefore, expresses strong cationic properties. This unique property of chitosan induced its application in nanoparticulate drug delivery systems using anionic drugs or DNA [17,18,19,20,21]. Furthermore, amine groups in the backbone of chitosan can be used to synthesize multifunctional nanoparticles [19,20,21]. For example, transferrin-conjugated chitosan can be complexed with anionic succinylated dextran MePEG for targeted DOX delivery against gliosarcoma cells [19]. Chitosan-based nanosheets enhanced active cellular internalization of drug [21].

DOX is a widely used anticancer agent for malignant lymphoma, leukemia, and soft tissue sarcoma. DOX produces free radicals, interferes with the action of DNA topoisomerase II, and damages the DNA of tumor cells [22]. Thus, it has anticancer efficacy through inhibition of the tumor cell proliferation [22]. However, DOX exhibits side effects such as suppression of bone marrow function, hair loss, stomatitis, nausea, vomiting, and tissue necrosis [23].

In this study, we synthesized IO nanoparticle-conjugated nanocomposites using MePEG-grafted chitosan copolymer for targeted delivery of DOX against colorectal carcinoma cells. DOX was also conjugated with a chitosan backbone using disulfide linkage. DOX- and IO-nanoparticle-conjugated ChitoPEG nanocomposites were characterized in an in vitro cell culture study and an in vivo animal tumor-xenograft model.

## 2. Results and Discussion

### 2.1. Synthesis and Characterization of ChitoPEG/DOX/IO Nanocomposites

Prior to the synthesis of nanocomposites, ChitoPEG copolymer was synthesized as previously described (Appendix A) [24]. Introduction of PEG in the backbone of the WSC chain may provide stealth properties to nanoparticles; i.e., hydrophilic PEG domain normally forms the outer shell of the nanoparticles and then protects nanoparticles from the attack of the reticuloendothelial system (RES) [1,2]. Furthermore, PEG is known to prolong blood circulation time and increase targeting efficacy [1,2]. As shown in 1H -NMR-spectra (Figure 1), a 3.2~3.8 ppm peak is estimated to be attributable to the ethylene protons, the methoxyl group of MePEG, and H-2 to H-6 of chitosan, respectively. In addition, the peaks between 4.4 and 4.6 ppm were estimated to be due to H-1 hydrogen of chitosan and 11–15 hydrogens of doxorubicin. Based on ^1^H NMR spectra, degree of substitution value was approximately 10.8 glucose unit/1PEG molecule. The observed peak near 2.5 ppm is assigned to originate from NH_2_ in chitosan. These results were obtained based on the literature and ^1^H-NMR analysis (Appendix A) of each Doxorubicin, Dithiodipropionicacid-NHS, DOX-CSSC-NHS, and Chitosan-g-PEG copolymer. DOX HCl reacted with dithiodipropionic acid NHS ester (Appendix A), and then DOX-CSSC-NHS conjugates were conjugated again with ChitoPEG copolymer as shown in Figure 1. DOX contents in the ChitoPEG/DOX conjugates were 8.1% (*w/w*), as shown in Table 1. Then, this was further conjugated with IO magnetic nanoparticles NHS-ester, as shown in Figure 1. This vigorously dialyzed against water to remove unreacted byproducts, and IO-nanoparticle-conjugated nanocomposites were separated by magnetism three times. When IO magnetic nanoparticles were attached to ChitoPEG/DOX conjugates, a broad spectrum was observed at ^1^H NMR spectra (Figure 1). This result might be due to IO nanoparticles in the nanocomposites having strongly pulled the backbone of the WSC and the cross-linked core of the nanoparticles. However, specific peaks of WSC and DOX were hardly confirmed.

Figure 2 shows the characterization of ChitoPEG/DOX/IO nanocomposites. As shown in Figure 2a and Table 1, particle sizes were increased when IO nanoparticles were conjugated with ChitoPEG/DOX conjugates and then fabricated ChitoPEG/DOX/IO nanocomposites. As shown in Figure 2b, the morphology of ChitoPEG/DOX/IO nanocomposites was spherical, and their sizes were between 100~300 nm. Interestingly, smaller IO nanoparticles less than 30 nm were also observed inside the larger particles, indicating that IO nanoparticles were successfully conjugated to ChitoPEG/DOX conjugates and then formed nanocomposites. These results indicate that ChitoPEG/DOX conjugates and IO nanoparticles not only were synthesized but also formed nanocomposites. Huang et al. also reported that IO-nanoparticle-embedded multifunctional nanocomposites have superior potential to diagnose cancer and aid in photothermal cancer therapy [25]. In their report, IO nanoparticles were embedded inside the multifunctional metal nanocomposites and can used to diagnose solid tumors in the animal tumor xenograft model. Our results also showed that IO nanoparticles were embedded inside the ChitoPEG/DOX/IO nanocomposites in TEM observations.

The intracellular GSH level in cancer cells is known to be elevated as opposed to those of an extracellular environment or normal cells [26]. These phenomena in cancer cells have encouraged many scientists to develop novel drug delivery systems based on the redox-responsive strategy [27,28,29,30]. Since the L-glutathione (GSH) level in cancer cells is normally elevated and GSH disintegrates the disulfide group, many scientists developed disulfide-linked polymeric micelles to improve DNA delivery and the acceleration of intracellular release of anticancer drugs [27,28,29]. From these points of view, we studied the redox-responsiveness of ChitoPEG/DOX/IO nanocomposites, as shown in Figure 3. Figure 3 shows the effect of the addition of GSH on the DOX release rate from nanocomposites. DOX release rate from ChitoPEG/DOX/IO nanocomposites was extremely slow at normal physiological conditions (GSH(−)), and less than 20% of the drug was released for four days. DOX release was minimized in this condition because DOX conjugated with the ChitoPEG copolymer via disulfide linkage, and this linkage is difficult to break in the absence of intracellular enzyme. This situation can be dramatically switched by the addition of the intracellular enzyme, GSH (GSH(+)). DOX release significantly increased when GSH was added to release media, indicating that ChitoPEG/DOX/IO nanocomposites have redox-responsiveness. Then, DOX release became sensitive to the redox environment. Furthermore, these results may provide the potential of intracellular targetability of ChitoPEG/DOX/IO nanocomposites. Park et al. reported that smart nanoparticles having disulfide linkages have a dual-targeting capacity [29]. They synthesized a hyaluronic acid (HA)-*b*-poly(DL-lactide-co-glycolide) (PLGA) block copolymer having disulfide linkage and smart nanoparticles. Using these, block copolymer can target CD44 receptors on the cancer cell surface together with redox-responsiveness. They also showed that their smart nanoparticles are able to target CD44 receptors in in vivo tumor models. Chu et al. also reported that dextran-chlorin e6 conjugates via disulfide linkage have dual sensitivity against colonic enzyme (dextranase) and GSH; i.e., chlorin e6 release was faster in the presence of dextranase or GSH enzyme [30]. In in vivo tumor models, their nanoparticles were absorbed by tumor tissue to a higher extent than chlorin e6 itself. Furthermore, they argued that these results induced improved anticancer activity in animal tumor models.

### 2.2. Anticancer Activity In Vitro

U87MG glioma cells (Figure 4a) and DOX-resistant CT26 colon cancer cells (Figure 4a) were employed to assess anticancer activity. As shown in Figure 4, cell viability was dose-dependently decreased according to DOX concentration in both DOX itself and ChitoPEG/DOX/IO nanocomposites. ChitoPEG/DOX/IO nanocomposites showed significantly higher anticancer activity against DOX-resistant CT26 cells; i.e., the viability of CT26 cells was higher than 80% with DOX treatment (2 µg/mL), while treatment with nanocomposites resulted in less than 60% viability. These results indicated that nanocomposites efficiently internalized into cancer cells and killed them.

Figure 5 shows the effect of external magnetic stimulus on the cellular uptake of nanocomposites. To achieve this stimuli-responsiveness, nanocomposites were exposed to cells for 1 h. As shown in Figure 5a, nanocomposites were extremely concentrated in the area of the magnet (Magnet (+)), and the fluorescence intensity in the cells was also much higher than in the absence of the magnet (Magnet (−)), indicating that ChitoPEG/DOX/IO nanocomposites have magnetic-sensitive drug delivery capacity. Furthermore, the viability of DOX-resistant CT26 cells was also decreased by magnetism. These results indicated that ChitoPEG/DOX/IO nanocomposites have superior stimuli-responsiveness over DOX delivery to cancer cells through a magnetism-sensitive manner.

### 2.3. In Vivo Animal Tumor-Bearing Mouse Study

CT26 cell-bearing mice were used to demonstrate magnetic-sensitive delivery of nanocomposites. Nanocomposites were intravenously administered via the tail vein of mice and a permanent magnet was held onto the solid tumor for 1 h, as shown in Figure 6.

As shown in Figure 6, strong fluorescence intensity was observed in the tumor tissue with magnet (+) while tumor tissue (Magnet (−)) showed little fluorescence intensity. Normal organs also represented negligible fluorescence intensity. These results indicated that ChitoPEG/DOX/IO nanocomposites have potential of magnetic-sensitive drug targeting and minimization of anticancer drug distribution into normal organs. Other investigators have also reported magnetic-sensitive drug delivery for cancer cells [31,32,33]. For example, an external magnetic field provides specific toxicity against lung cancer cells with minimized toxicity against normal fibroblast cells [29]. Furthermore, magnetic responsive nanoparticles are also known to overcome obstacles of drug penetration and enhance drug delivery into deep tissues or organs [32,33,34].

## 3. Materials and Methods

### 3.1. Materials

Water-soluble chitosan (WSC) (molecular weight (*M*_W_): 7000 g/mol, deacetylation degree >97%) was purchased from Kittolife Co. Ltd. (Seoul, Korea). Doxorubicin hydrochloride (DOX), *N*-succinimidyl ester functionalized iron oxide(II,III) magnetic nanopowder (IO-NHS), 3,3′-dithiodipropionic acid di(*N*-hydroxysuccinimide ester) (CSSC-NHS), 3-(4,5-dimethyl-2-thiazolyl)-2,5-diphenyl-2H-tetrazolium bromide (MTT), dimethyl sulfoxide (DMSO), and triethylamine (TEA) were purchased from Sigma Chem. Co. Ltd. (St. Louis, MO, USA). Methoxy poly(ethylene glycol) *N*-hydroxysuccinimide (MePEG-NHS) (*M*_W_ = 5000 g/mol) was purchased from Sunbio Co. Inc. (Seoul, Korea). Dialysis membrane (*M*_W_ cut-off-size: 8000 g/mol) was purchased from Spectrum Lab. Inc. (Gardena, CA, USA). Cell culture media and related materials were purchased from Invitrogen (New York, NY, USA). All other chemicals and organic solvents were extra-pure grade.

### 3.2. Synthesis DOX-Conjugated ChitoPEG Magnetic (ChitoPEG/DOX/IO) Nanocomposites

Synthesis of copolymer was performed as reported previously [22]. Briefly, 180 mg of WSC dissolved in 5 mL of deionized water was mixed with 10 mL DMSO. To this solution, MePEG-NHS (500 mg) was added and then magnetically stirred for 2 days. This solution was introduced into a dialysis membrane (*M*_W_ cut-off size, 8000 g/mol) and then dialyzed against 1 L deionized water for 3 days to remove unreacted MePEG and byproducts. To avoid saturation, water was exchanged at 3 h intervals. Resulting solution lyophilized for 3 days and a yellowish powder was obtained. Lyophilized solid was further purified by precipitation into excess chloroform and then dried under a vacuum for 2 days. After that, this solid (ChitoPEG copolymer) was restored in 4 °C until it was used for synthesis or analysis.

ChitoPEG-DOX conjugates: 21.5 mg of DOX HCl was dissolved in 5 mL of DMSO with one drop of TEA, and then 15 mg of CSSC-NHS was added (DOX.HCl/CSSC-NHS = 1/1 molar ratio). This solution was magnetically stirred for 6 h to make DOX-CSSC-NHS conjugates. A total of 193 mg of ChitoPEG copolymer dissolved in 5 mL deionized water was mixed with 10 mL of DMSO. This solution was then added to the DOX-CSSC-NHS solution. Mixed solution was magnetically stirred for 48 h. After that, this solution was dialyzed against deionized water to remove unreacted DOX-CSSC-NHS and byproducts for 2 days with exchange of water at 3 h intervals. Then, dialyzed solution was lyophilized for 3 days, and ChitoPEG-DOX conjugates were observed.

ChitoPEG/DOX/IO nanocomposites: 220 mg of ChitoPEG-DOX conjugates was distributed in 10 mL phosphate-buffered saline (PBS, 0.01 M, pH 7.4) by sonication (1 s × 30), and then 10 mL of DMSO was added. To this solution, 20 mg of IO-NHS was added and then agitated at 200 rpm using a shaker incubator at 200 rpm (20 °C, HB-201SF, Hanbaek Sci. Co., Seoul, Korea) for 2 days. This solution was also dialyzed against water for 2 days and then lyophilized for 3 days. Final lyophilized solid was obtained as ChitoPEG-DOX/IO nanocomposites.

Empty nanocomposites: 200 mg of ChitoPEG copolymer in 10 mL PBS was mixed with 10 mL DMSO. Then, 20 mg of IO-NHS was added following agitation at 200 rpm using shaker incubator at 200 rpm for 2 days. This solution was also dialyzed against water for 2 days and then lyophilized for 3 days.

### 3.3. Characterization of Magnetic Nanocomposites

^1^H NMR spectra (500 mHz superconducting Fourier transform (FT)-NMR spectrometer, Varian Unity Inova 500 MHz NB High-Resolution FT NMR; Varian Inc., Santa Clara, CA, USA) was used to monitor the synthesis of conjugates. To measure the ^1^H NMR spectra, conjugates were dissolved in D_2_O, DMSO, or D_2_O/DMSO mixtures.

A transmission electron microscope (TEM) (H-7600, Hitachi Instruments Ltd., Tokyo, Japan) was used to observe the morphology of ChitoPEG/DOX/IO nanocomposites. ChitoPEG/DOX/IO nanocomposites in water (nanoparticle weight 1 mg/mL) were dropped onto a carbon film-coated copper grid and dried at room temperature overnight. TEM observation was carried out at 80 kV.

Particle size of ChitoPEG/DOX/IO nanocomposites was analyzed with Nano-ZS at room temperature (Malvern, Worcestershire, UK) (nanoparticle content in water was less than 0.1 wt.%).

### 3.4. Drug Release Study

Five milligrams of nanocomplexes were reconstituted into 5 mL phosphate-buffered saline (PBS, pH 7.4, 0.01 M) with sonication (1 s × 10). Then, this solution was introduced into a dialysis membrane and put into a conical tube with 40 mL of PBS (0.01 M, pH 7.4). Release study was performed at an agitation speed of 100 rpm and temperature of 37 °C. PBS in the tube was harvested and replaced with fresh PBS at specific time intervals. The harvested media were used to measure the concentration of released DOX using ultraviolet spectrophotometer (UV-1601, Shimadzu Co. Ltd., Osaka, Japan) at 489 nm. Five milligrams of empty nanocomposites were also reconstituted into PBS and the released media were used as a blank test. All experiments were separately repeated three times, and the results were expressed as mean ± S.D.

### 3.5. Cell Culture Study

CT26 mouse colorectal carcinoma cells were obtained from the Korean Cell Line Bank Co. Ltd. (Seoul, Korea). CT26 cells were cultured with RPMI1640 media supplemented with 10%FBS and 1% antibiotics. DOX-resistant CT26 cells were prepared as described previously [17]. CT26 cells were exposed to DOX (0.001 µg/mL) for 3 h and then replaced with fresh media. Cells were further cultured in a CO_2_ incubator (37 °C) for 2 days. This procedure was repeated 3 times and then cell exposure to DOX concentration was gradually increased to 0.1 µg/mL.

Cell cytotoxicity test was performed as follows: 3 × 10^4^ CT26 cells seeded in 96 wells were exposed to free DOX, empty nanocomposites, or ChitoPEG/DOX/IO nanocomposites. DOX (Free DOX treatment) was dissolved in DMSO and then diluted with serum-free RPMI1640 media (final concentration of DMSO: 0.5%, *v/v*). Empty nanocomposites or ChitoPEG/DOX/IO nanocomposites reconstituted into PBS were also diluted with serum-free media. Cells were further incubated in a 5% CO_2_ incubator at 37 °C for 1 or 2 days. After that, 30 µL of MTT reagent (5 mg/mL in PBS) was added to 96 wells and further incubated in a 5% CO_2_ incubator at 37 °C. Four hours later, supernatants were removed and DMSO (100 µL) was added in 96 wells to measure absorbance at 570 nm using a microplate reader (Infinite M200 pro multimode microplate readers, Tecan Trading AG Inc., Männedorf, Switzerland). The results were the average of 8 wells and expressed as mean ± SD.

To observe morphology of cells, 1 × 10^5^ cells were seeded in 6 wells with glass cover. DOX or ChitoPEG/DOX/IO nanocomposites were exposed to cells. One hour later, cells were washed with PBS, fixed with 10% paraformaldehyde solution for 10 min, and immobilized with immobilization solution (Immunomount, Thermo Electron Co. Pittsburgh, PA, USA). Fluorescence microscope (Eclipse 80i; Nikon, Tokyo, Japan) was used to observe images of treated cells.

### 3.6. In Vivo Fluorescence Imaging

For tumor-xenograft model of small animals, male BALb/C nude mice were used (20–25 g, 4~5 weeks old). A total of 1 × 10^6^ CT26 cells were subcutaneously (s.c.) implanted in the back of mice. When the diameter of solid tumor reached approximately 5 mm, mice were used for experiment. Mice were left with free access to food and water.

CT26-bearing mice were used to observe in vivo imaging. ChitoPEG/DOX/IO nanocomposites (DOX dose: 5 mg/kg, injection volume: 100 µL) in PBS were administered intravenously (i.v.) via tail vein of mice. To study stimuli-sensitivity, a neodymium magnet was connected to the solid tumor for 60 min. A small animal imaging instrument (Maestro^TM^ 2, Cambridge Research & Instrumentation, Inc., Hopkinton, MA, USA) was used to observe whole body images of the mice, and, after that, mice were sacrificed to observe fluorescence images for each organ. The target organ delivery ability of anticancer drugs not only maximizes the effect of the drug but also minimizes the distribution of the drug outside the tumor tissue, thereby reducing the side effects of the drug. Vascular endothelial cells distributed in tumor tissues are widely used as pathways for polymer drug delivery systems such as nanoparticles. Polymer-based drug carriers can deliver a large amount of drug to tumor tissue and avoid the recognition of p-glycoprotein, which plays an important role in drug excretion, thereby increasing the efficiency of anticancer drugs. As a result of this study, it is estimated that ChitoPEG/DOX/IO nanoparticles can be efficiently integrated into tumor tissues and can reduce the side effects of drugs.

### 3.7. Animal Tumor Xenograft Study

To study magnetic-responsive anticancer activity of nanoparticles, nanoparticles or DOX were given to CT26-bearing mice as described above. DOX alone or nanoparticles were intravenously (i.v.) administered via the tail vein of the mice when solid tumor reached about 4~5 mm diameter. Injection volume was 100 µL. A permanent magnet (neodymium magnet) was held onto one side (right side) of the solid tumor for 1 h. Four weeks later, all mice were sacrificed, and tumor tissues were harvested to compare tumor weight.

Statistical analysis of the results was performed using a *t*-test with *p* < 0.01 as the minimal level of significance.

## 4. Conclusions

Nanocomposites were synthesized using ChitoPEG copolymer for dual-sensitivity against the intracellular GSH of cancer cells and a magnetic stimulus. To provide GSH-sensitivity, DOX was conjugated in the backbone of the WSC chain via disulfide linkage and, to give magnetic-responsiveness, IO nanoparticles were also conjugated. ChitoPEG/DOX/IO nanocomposites showed small particle sizes around 150 nm, while the diameter of nanoparticles of ChitoPEG/DOX conjugates was less than 100 nm. DOX release rate was changed by the addition of GSH, indicating that nanocomposites have sensitivity against intracellular enzymes of cancer cells. Furthermore, nanocomposites were dramatically concentrated in the external magnetic field and efficiently internalized into cells in the cell culture experiment. Nanocomposites also showed higher anticancer activity in the presence of an external magnetic stimulus. In the animal tumor-xenograft study, nanocomposites preferentially targeted tumor tissues by the use of an external magnetic stimulus and showed increased fluorescence intensity. We suggested ChitoPEG/DOX/IO nanocomposites as a superior candidate for anticancer drug targeting.

## Figures and Tables

**Figure 1 ijms-22-13169-f001:**
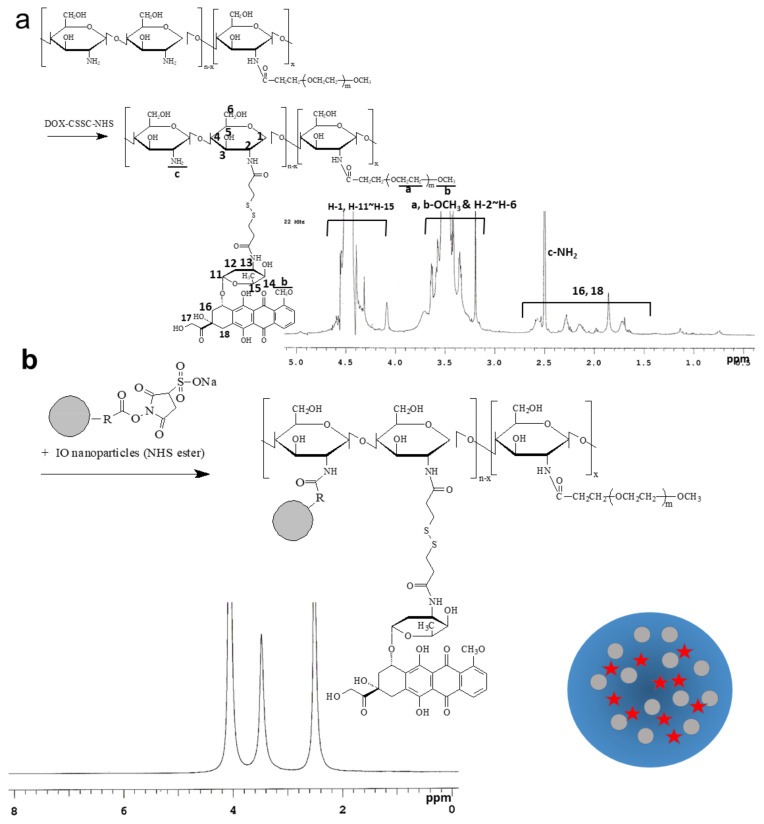
(**a**) Synthesis scheme and (**b**) ^1^H NMR spectra of DOX- and IO-nanoparticle-conjugated ChitoPEG copolymer (abbreviated as ChitoPEG /DOX/IO nanocomposites).

**Figure 2 ijms-22-13169-f002:**
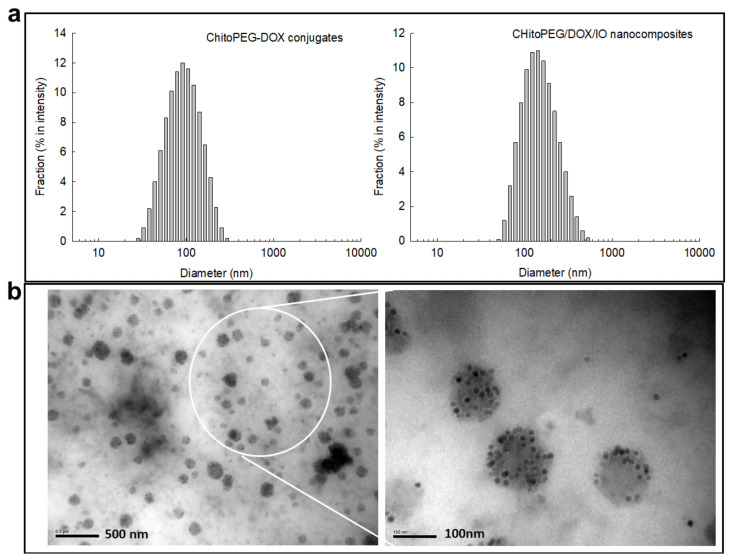
Characterization of ChitoPEG/DOX/IO nanocomposites. (**a**) Particle size distribution of ChitoPEG/DOX conjugates or ChitoPEG/DOX/IO nanocomposites. (**b**) TEM images of ChitoPEG/DOX/IO nanocomposites.

**Figure 3 ijms-22-13169-f003:**
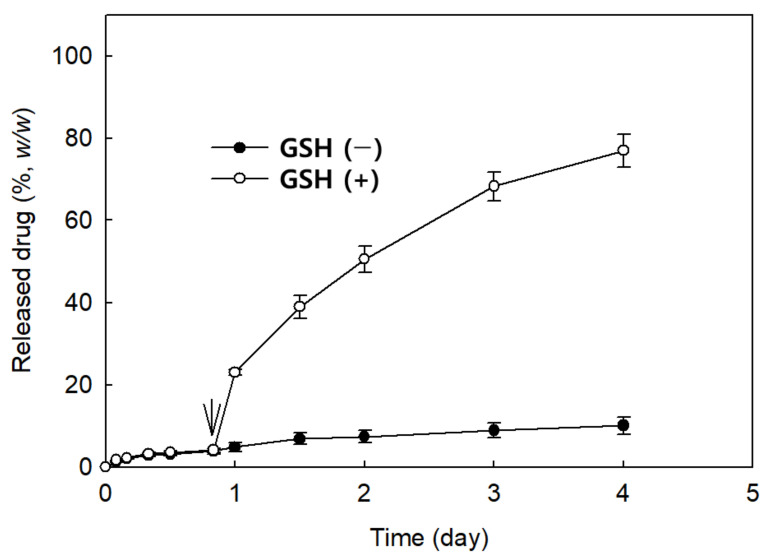
DOX release from ChitoPEG/DOX/IO nanocomposites.

**Figure 4 ijms-22-13169-f004:**
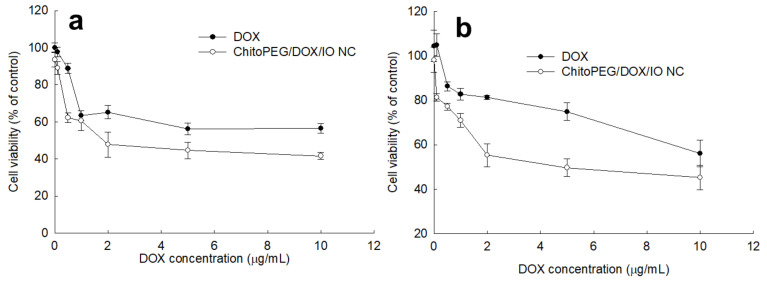
Anticancer activity of DOX and ChitoPEG/DOX/IO nanocomposites against U87MG glioma cells (**a**) and DOX-resistant CT26 mouse colorectal carcinoma cells (**b**). NC = nanocomposites.

**Figure 5 ijms-22-13169-f005:**
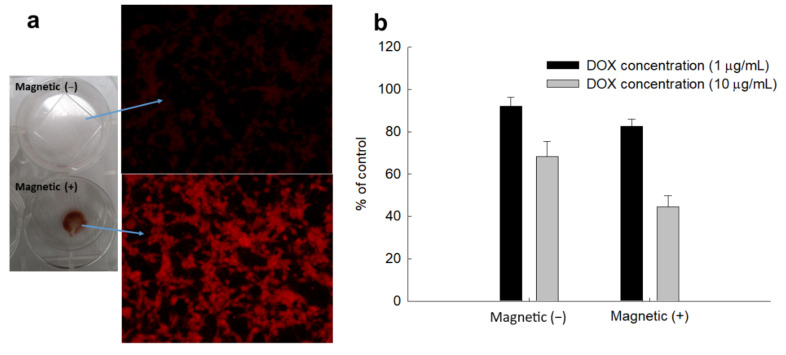
Magnetically-responsive delivery of ChitoPEG/DOX/IO nanocomposites against DOX-resistant CT26 mouse colorectal carcinoma cells in vitro. (**a**) Cell images of magnetic responsiveness of nanocomposites. (**b**) Viability of CT26 cells. To study magnetic-responsiveness, nanocomposites treated to cells for 1h and then cells were washed to observe.

**Figure 6 ijms-22-13169-f006:**
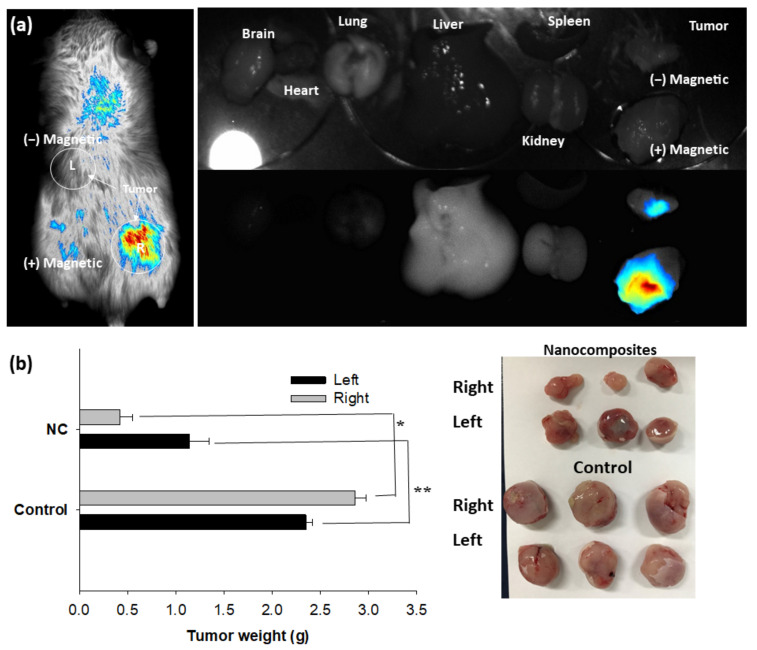
(**a**) Magnetically-responsive tumor targeting of ChitoPEG/DOX/IO nanocomposites against in vivo animal tumor model of DOX-resistant CT26 mouse colorectal carcinoma cells. (**b**) Tumor weight comparison of CT26 cell-bearing mice. 100 µL of ChitoPEG/DOX/IO nanocomposites aqueous solution (dose: 10 mg DOX/kg mice) were intravenously injected via tail vein of mice. 3 weeks later, mice were sacrificed and tumor tissue was separated to measure weight. The results were mean ± S.E. from 3 mice. *, ** *p* < 0.01.

**Table 1 ijms-22-13169-t001:** Characterization of ChitoPEG/DOX conjugates and ChitoPEG/DOX/IO) nanocomposites.

	Drug Contents (%, *w/w*)			Particle Size (nm) ^b^
	Theoretical ^a^	Experimental ^a^	Conjugation Yield	
ChitoPEG/DOX conjugates	9.1	8.1	89	81.5
ChitoPEG/DOX/IO nanocomposites	−	7.5	−	148.9

^a^ Theoretical drug contents = [(Feeding weight of DOX/(Weight of ChitoPEG copolymer + Feeding weight of DOX)] × 100. Experimental drug contents (ChitoPEG/DOX conjugates) = [Measured weight of DOX from the ChitoPEG/DOX conjugates/(Weight of ChitoPEG copolymer + Feeding weight of DOX)] × 100. Experimental drug contents (ChitoPEG/DOX/IO nanocomposites) = [(Measured weight of DOX from the ChitoPEG/DOX/IO nanocomposites)/(Weight of ChitoPEG copolymer + weight of IO nanoparticles + Feeding weight of DOX)] × 100. ^b^ Particle sizes were intensity fractions.

## Data Availability

Not applicable.

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
