# Peer review of "Stimuli-Responsive Drug Delivery of Doxorubicin Using Magnetic Nanoparticle Conjugated Poly(ethylene glycol)-g-Chitosan Copolymer"

_ijms, 2021, doi:10.3390/ijms222313169_

Round 1

Reviewer 1 Report

The manuscript presents the results of the studies on complex nanostructured systems based on chitosan and containing a selected drug (doxorubicin) as well as magnetic nanoparticles.

 Unfortunately, the overall quality of the manuscript is poor especially with respect of the presentation of the results. It should be improved prior further evaluation.

Some examples:

  1. 3 line 98-100 – the explanation of the lack of clear NMR spectrum of the final system containing magnetic nanoparticles is unrealistic. The influence of magnetic nanoparticles on the relaxation of protons in NMR should be taken into account and other methods than NMR should be used for characterization of the system.

Table 1 – it is unclear what “Title 3” means.

Figure 2 – the caption is totally wrong. The reference to Figure 2 in the text is also wrong. Figure 2 seems to present TEM images of the systems containing magnetic nanoparticles while in the text it refers also to the polymers without the nanoparticles

Abstrac: “diameter of nanocomposites” – nanocomposites consist of dispersed nanoparticles in a matrix thus it is improper to present a size of nanocomposites as stated in the abstract. The word “nanocomposite” may not be adequate to described the system composed of polymer-based nanoparticles containing smaller magnetic nanoparticles.

The language of the manuscript requires significant improvement due to numerous grammatical errors that make also some parts of the text difficult to understand or ambiguous.

Author Response

The manuscript presents the results of the studies on complex nanostructured systems based on chitosan and containing a selected drug (doxorubicin) as well as magnetic nanoparticles.

 Unfortunately, the overall quality of the manuscript is poor especially with respect of the presentation of the results. It should be improved prior further evaluation.

Some examples:

  1. 3 line 98-100 – the explanation of the lack of clear NMR spectrum of the final system containing magnetic nanoparticles is unrealistic. The influence of magnetic nanoparticles on the relaxation of protons in NMR should be taken into account and other methods than NMR should be used for characterization of the system.

Answer) Thanks for your comment. At this moment, we have some difficulties to find other kind of methods instead of NMR because we have no device in my department. Therefore, we indicated each peaks of compound, copolymer and nanocomplexes in the Figure 1, Figure S1 and Figure S2. Furthermore, we explained and discussed more as follows:

2.1. Synthesis and characterization of DOX/IO nanoparticle-conjugated ChitoPEG nanocomposites

Prior to synthesis of nanocomposites, ChitoPEG copolymer was synthesized as previously described (Figure s1) [24]. Introduction of PEG in the backbone of WSC chain may provide stealth properties to nanoparticles, i.e. hydrophilic PEG domain normally forms outer-shell of the nanoparticles and then protects nanoparticles from attack of reticuloendothelial system (RES) [1,2]. Furthermore, PEG is known to prolong blood circulation time and to increase targeting efficacy [1,2]. As shown in 1H -NMR-spectra (Figure 1), 3.2~3.8 ppm peak is estimated to be attributable to the ethylene protons, the methoxyl group of MePEG, and H-2 to H-6 of chitosan, respectively. In addition, the peaks between 4.4-4.6ppm were estimated to be due to H-1 hydrogen of chitosan and 11-15 hydrogens of doxorubicin. Based on 1H NMR spectra, degree of substitution value was approximately 10.8 glucose unit/1PEG molecule. These results were obtained based on the literature and 1H-NMR analysis (Figure s1, s2) of each Doxorubicin, Dithiodipropionicacid-NHS, DOX-CSSC-NHS, Chitosan-g-PEG copolymer. DOX HCl was reacted with dithiodipropionic acid NHS ester (Figure s2) and then DOX-CSSC-NHS conjugates were conjugated again with ChitoPEG copolymer as shown in Figure 1. DOX contents in the ChitoPEG/DOX conjugates 8.1 % (w/w) as shown in Table 1. Then, this was further conjugated with IO magnetic nanoparticles NHS-ester as shown in Figure 1. This vigorously dialyzed against water to remove unreacted byproducts and IO nanoparticle-conjugated nanocomposites were separated by magnetism three times. When IO magnetic nanoparticles were attached to ChitoPEG/DOX conjugates, broad spectrum was observed at 1H NMR spectra (Figure 1). This result might be due to that IO nanoparticles in the nanocomposites strongly pulled backbone of the WSC and cross-linked core of the nanoparticles. Then, specific peaks of WSC and DOX were hardly confirmed.

Figure 1. (a) Synthesis scheme and (b) 1H NMR spectra of DOX and IO nanoparticle conjugated ChitoPEG copolymer (abbreviated as ChitoPEG/IO/DOX-CSSC conjugates).

In supplementary materials

Figure s1 shows the synthesis scheme of ChitoPEG copolymer and 1H NMR spectra. As shown in Figure s1, 3.2~3.8 ppm peak in 1H -NMR-spectra is estimated to be attributable to the ethylene protons, the methoxyl group of MePEG, and H-2 to H-6 of chitosan, respectively(Figure s1). In 1H-NMR of Doxorubicin (Figure s2), characteristic peaks due to hydrogens 7, 8, and 9 in the benzene ring were observed at 7.5~8.0ppm. In dithiodipropionicacid–NHS 1H-NMR, the peaks at 5.4-5.9 ppm for hydrogens 19 and 2.6-3.2 for hydrogens 20 were observed, respectively. In the analysis of 1H-NMR spectra of DOX-CSSC-NHS, a sharp peak due to the 10-methoxy group of doxorubicin was observed at 3.4 ppm, and a peak due to hydrogens 11-15 was observed between 3.0 and 4.5 ppm. Based on 1H NMR spectra, degree of substitution value was approximately 10.8 glucose unit/1PEG molecule. DOX HCl was reacted with dithiodipropionic acid NHS ester (Figure s2).

Figure s1. (a) Synthesis scheme and (b) 1H NMR spectra of ChitoPEG graft copolymer.

Figure s2. (a) Synthesis scheme and (b) 1H NMR spectra of DOX-dithiodipropinic acid N-hydroxysucinimide ester.

Table 1 – it is unclear what “Title 3” means.

Answer) Thanks for your comment. According to your comment, we corrected the Table 1 as follows.

Drug contents (%, w/w)

Particle size (nm) b

Theoretical a

Experimental a

Conjugation yield

ChitoPEG/DOX conjugates

9.1

8.1

89

81.5

ChitoPEG/DOX/IO nanocomposites

-

7.5

-

148.9

Table 1. Characterization of magnetic nanoparticle and DOX-conjugated ChitoPEG (ChitoPEG/IO/DOX) nanocomposites.

a Theoretical drug contents = [(Feeding weight of DOX/(Weight of ChitoPEG copolymer + Feeding weight of DOX)]* 100.

Experimental drug contents (ChitoPEG/DOX conjugates) = [Measured weight of DOX from the ChitoPEG/DOX conjugates/(Weight of ChitoPEG copolymer + Feeding weight of DOX)]* 100.

Experimental drug contents (ChitoPEG/DOX/IO nanocomposites) = [(Measured weight of DOX from the ChitoPEG/DOX/IO nanocomposites)/(Weight of ChitoPEG copolymer + weight of IO nanoparticles + Feeding weight of DOX)] * 100.

b Particle sizes were intensity fractions.

Figure 2 – the caption is totally wrong. The reference to Figure 2 in the text is also wrong. Figure 2 seems to present TEM images of the systems containing magnetic nanoparticles while in the text it refers also to the polymers without the nanoparticles

Answer) Thanks for your comment. According to your comment, we corrected these mistakes and revised Figure 2as follows:

Figure 2 showed the characterization of nanoparticles of ChitoPEG/DOX/IO nanocomposites. As shown in Figure 2(a) and table 1, particle sizes were increased when IO nanoparticles were conjugated with ChitoPEG/DOX conjugates and then fabricated ChitoPEG/DOX/IO nanocomposites. As shown in Figure 2(b), the morphology of ChitoPEG/DOX/IO nanocomposites were spherical and their sizes were between 100 ~ 300 nm. Especially, smaller IO nanoparticles less than 30 nm also observed inside the larger particles, indicating that IO nanoparticles were successfully conjugated to ChitoPEG/DOX conjugates and then formed nanocomposites. These results indicated that ChitoPEG/DOX conjugates and IO nanoparticles were not only synthesized but also formed nanocomposites. Huang et al. also reported that IO nanoparticle-embedded multifunctional nanocomposites have superior potential to diagnose and photothermal cancer therapy [25]. In their report, IO nanoparticles was embedded inside the multifunctional metal nanocomposites and can be used to diagnose solid tumor in the animal tumor xenograft model. Our results also showed that IO nanoparticles were embedded inside the ChitoPEG/DOX/IO nanocomposites in TEM obserbvations.

Figure 2. Characterization of ChitoPEG/DOX/IO nanocomposites. (a) Particle size distribution of ChitoPEG/DOX conjugates or ChitoPEG/DOX/IO nanocomposites. (b) TEM images of ChitoPEG/DOX/IO nanocomposites

Abstrac: “diameter of nanocomposites” – nanocomposites consist of dispersed nanoparticles in a matrix thus it is improper to present a size of nanocomposites as stated in the abstract. The word “nanocomposite” may not be adequate to described the system composed of polymer-based nanoparticles containing smaller magnetic nanoparticles.

Answer) Thanks for your comment. According to your comment, we delete the diameter of the nanocomposites because it is nor proper to present particle sizes as you indicated. Furthermore, we discussed more about the nanocomposites in the manuscript.

Huang et al. also reported that IO nanoparticle-embedded multifunctional nanocomposites have superior potential to diagnose and photothermal cancer therapy [25]. In their report, IO nanoparticles was embedded inside the multifunctional metal nanocomposites and can be used to diagnose solid tumor in the animal tumor xenograft model. Our results also showed that IO nanoparticles were embedded inside the ChitoPEG/DOX/IO nanocomposites in TEM obserbvations.

  1. Huang, Y.; Wei, T.; Yu, J.; Hou, Y.; Cai, K.; Liang, X. Multifunctional metal rattle-type nanocarriers for MRI-guided photothermal cancer therapy. Mol. Pharm. 2014, 11, 3386-3394.

The language of the manuscript requires significant improvement due to numerous grammatical errors that make also some parts of the text difficult to understand or ambiguous.

Answer) Thanks for your comment. According to your comment, we corrected the English expressions and improved the English expression. I appreciated again.

Reviewer 2 Report

The paper is well written and discussed. There are minor changes required for the acceptance i.e. higher resolution for the pictures. Table 1 also contains "title 3" which is mostly an error. Authors should also explain how to obtain theoretical and experimental drug content. After these adjustment, I recommend publication of this paper

Author Response

Response ro Reviewer 2’s comment

The paper is well written and discussed. There are minor changes required for the acceptance i.e. higher resolution for the pictures. Table 1 also contains "title 3" which is mostly an error. Authors should also explain how to obtain theoretical and experimental drug content. After these adjustment, I recommend publication of this paper

Answer) Thanks for your kind review. I appreciated your comment. According to your comment, we explain how to measure theoretical and experimental drug contents. Furthermore, we corrected the title 3 in Table 3.

Drug contents (%, w/w)

Particle size (nm) b

Theoretical a

Experimental a

Conjugation yield

ChitoPEG/DOX conjugates

9.1

8.1

89

81.5

ChitoPEG/DOX/IO nanocomposites

-

7.5

-

148.9

Table 1. Characterization of magnetic nanoparticle and DOX-conjugated ChitoPEG (ChitoPEG/IO/DOX) nanocomposites.

a Theoretical drug contents = [(Feeding weight of DOX/(Weight of ChitoPEG copolymer + Feeding weight of DOX)]* 100.

Experimental drug contents (ChitoPEG/DOX conjugates) = [Measured weight of DOX from the ChitoPEG/DOX conjugates/(Weight of ChitoPEG copolymer + Feeding weight of DOX)]* 100.

Experimental drug contents (ChitoPEG/DOX/IO nanocomposites) = [(Measured weight of DOX from the ChitoPEG/DOX/IO nanocomposites)/(Weight of ChitoPEG copolymer + weight of IO nanoparticles + Feeding weight of DOX)] * 100.

b Particle sizes were intensity fractions.

Round 2

Reviewer 1 Report

The authors generally responded to the comments.